# Investigation of the Condition of the Gold Electrodes Surface in a Plasma Reactor

**DOI:** 10.3390/ma12132137

**Published:** 2019-07-03

**Authors:** Sebastian Gnapowski, Elżbieta Kalinowska-Ozgowicz, Mariusz Śniadkowski, Aleksandra Pietraszek

**Affiliations:** Fundamentals of Technology Faculty, Lublin University of Technology, 20-618 Lublin, Poland

**Keywords:** gold, surface of electrodes, ozone, plasma, raids layer

## Abstract

During the long-term operation of a plasma reactor, decreases in plasma concentration were noticed despite the constant maintenance of all parameters. One of the factors was the decrease in the nitrogen content on the electrode surface; in order to eliminate it, the supply voltage was increased up to 11 kV. Another decisive factor in the plasma concentration decrease was the oxidation of the electrode surface. These effects were studied using two electrodes: a gold one and a copper one coated with a 10 μm thick layer of galvanized gold. In the experiment with the gold coated electrode, a large decrease in plasma concentration was observed. High-energy electrons knocked out the gold atoms from the electrode; as a result, the gold atoms evaporated and the raids layers formed. After the electrodes had been in operation for a month, metallographic analyzes were carried out, the results of which are described in this paper.

## 1. Introduction

A rotating electrode was coated with gold and a high voltage was applied to it. The speed of the rotating electrode was varied from 0 to 800 rpm by a variable speed motor. The dielectric barrier covered by a mesh electrode was a glass tube of length 110 mm, diameter 15 mm and thickness 1.25 mm. The outer mesh electrode made from copper wires with diameters of 0.1 mm was grounded. The size of the copper mesh electrode was 0.2 mm square. Also, 99.5% oxygen gas regulated by a digital mass flow controller was fed at a gas flow rate ranging from 0.5 l/min to 2 l/min. The applied voltage and its frequency were set at 9–10 kV and about 12 kHz, respectively. Figure 1 shows a schematic diagram of our ozonizer. The discharge gap distance was 1.1 mm and the discharge length along the reactor was 100 mm. The gas temperature at the outlet of the ozonizer was measured during the experiments. The measured data was saved every day in a computer. All experiments were carried out at atmospheric pressure in oxygen at around room temperature (15~30 °C). In our case, the cooling water temperature was 10 °C. 

The factor that necessitated the investigation of electrodes used in the plasma reactor was the decreasing performance of the device during ozone production while maintaining constant process parameters [1,2,3,4]. The only noticeable change during the reactor operation was the change in the surface condition of the electrodes used (Figure 2) and the concentration of the ozone produced (Figure 3). Ozone concentration decreased with working time day by day, and during experiments with the gold-coated copper electrode, ozone comcentration decreased to close to zero [5,6,7,8]. The plasma reactor was operated at an elevated voltage to eliminate the ozone zero phenomena effect. At higher voltages, nitrogen absorbed on the surface of the electrode is released and takes part in the ozone formation process. This is due to Einstein’s theory, because when an electron collides with an oxygen molecule, two oxygen atoms are created plus the energy that nitrogen must absorb, otherwise no ozone is formed [9,10,11,12]. 

After one month of operation, the electrode made of copper coated with galvanized gold had a matte surface with a dark coating indicating the presence of a layer of oxidation products—Figure 2b. Gold was minted by energetic plasma electrons and evaporated. The gold electrode showed no macroscopic changes in polish and surface coloring—Figure 2a [13,14,15,16]. A surface condition is most important [17,18,19,20,21,22,23].

The aim of the experiment was to determine the structural effects of the chemical corrosion process of the surface of rotating electrodes operating in an environment of dynamic oxygen flow and ozone in an electric field of a defined intensity [24,25,26,27,28,29,30].

The scope of work includes:(1)preparation of research material;(2)metallographic research of the electrode material in the scope of microstructure and geometry, construction and phase composition of the raids layers on their surface;(3)identification of interstitial phases in the raids layers using X-ray diffraction and scanning electron microscopy;(4)microanalysis of the chemical composition of the raids layers using the EDX (Energy Dispersive X-Ray) attachment in the scanning electron microscope.

## 2. Materials and Methods 

The experimental material consisted of electrodes made of a solid copper coated electroplated with 24 carat gold of 10 μm thickness and 10 mm solid gold. The chemical composition of electrode materials is given in Table 1. 

Experimentally tested electrodes in the form of rollers with a length of 100 mm and a diameter of 10 mm had an M6 internal thread on one side, and on the other, a hole Ø 6 mm with milling for the clamping wedge on the drive shaft—Figure 4.

To determine the structure of the materials used for electrodes and to comprehensively evaluate the metallographic effects of chemical corrosion of the surface of electrodes occurring during the plasma production process, experiments were carried out using light microscopy, scanning electron microscopy and X-ray phase abanation.

Metallographic microscopic investigations were carried out on the material cut off from the electrodes after their 30-day operation. The cut material was embedded in a self-hardening resin and ground mechanically on aqueous abrasive papers. Ground miscrosections were polished mechanically using diamond pastes with different granulation and the gold electrode was purified in a reagent, a compound of sulfuric and nitric acid. Metallographic observations and grain surface measurements were performed using the OLIMPUS GX71 (OLIMPUS Corporation, Tokyo, Japan) reflecting optical microscope using a computer image analysis system using zoom from 100 to 2000×.

X-ray researches performed with the diffractometric method included a qualitative X-ray phase analysis of the electrode surface after 30 days of use in a plasma generator (plasma reactor designed and made by the author Gnapowski S.). Electrode X-ray investigations were carried out using an Empyrean X-ray diffractometer from PANalytical. The investigations were carried out using a parallel beam technique in configuration with a Pixcel cobalt anode CoλKα anode at 35 kV and an anode current of 25 mA. The X-ray qualitative phase analysis was carried out in the range of 2θ angles from 10° to 100° corresponding to the interplanar distances between 1.027 and 0.1168 nm. The experimental conditions were ensured in which the resulting diffractograms represent the material of the surface layer of the research electrodes. The identification of the phase composition of the layers formed on the surface of the research electrodes was based on the base of the International Center for Diffraction Data PDF (Powder Diffraction File) - 4 + version 2015. 

Research using a scanning microscope was carried out to assess the structure of the electrodes and the chemical composition in micro-areas of raids layers formed as a result of the oxidation process. Observations were made on metallographic microsections using the ZEISS SUPRA 35 electron microscope. The investigation used a side and intra-lens detector, using secondary electron detection. The chemical composition of the micro-areas of the raids layers was determined using the EDX system. The observation was carried out at a 20 kV accelerating voltage using a zoom up to 2000×.

## 3. Results

At first, the gold layer of the copper electrode’s surface was investigated. Structural testing of the electrodes was carried out by metallographic observation using light microscopy and X-ray microanalysis, as well as phase analysis using scanning electron microscopy. The subject of metallographic research was the assessment of the structure of the electrode material as well as the geometry of the raids layers on the surface of the electrodes after a month of operation in the ozonator. The determination of the phase composition of the raids layers and their chemical composition in the micro-areas required the use of X-ray diffraction and the XAM method (X-ray analysis method) with high resolution. The results of metallographic observations using a light microscope are included in micro photographs—Figure 5 and Figure 6. X-ray analysis results were presented on the diffraction patterns—Figure 7 and Figure 13, and microanalysis made with the use of EDX, scanning electron microscopy on micro photographs (Figure 8, Figure 9, Figure 10 and Figure 11), and in Table 1 and Table 2.

Based on the results of metallographic research, it was found that the layers occurring on electrodes subjected to a corrosive, aggressive environment of oxygen and ozone during a month of operation in the plasma reactor, whose thickness did not exceed 10 μm, were characterized by the geometry corresponding to the coating layers.

A copper electrode with a galvanically-applied layer of gold is characterized after a month’s use by a structure of large recrystallized copper grains with straight lines and a discontinuous layer of gold—Figure 5. Between the galvanic layer of gold and copper there are areas with no adhesion, mainly at the points where the boundaries between Cu and Au layers meet—Figure 8. The structure of the electroplated Au layer and the occurring discontinuities may be the reason for the ejection of gold atoms from the galvanic layer onto the plasma reactor housing during its operation, formation of discontinuities in the layer, progress of the surface oxidation process, and, as a result, a decrease in ozone concentration during the work of the plasma reactor. Observations using the scanning electron microscope revealed the occurrence of oxidation effects in the form of a very thin raid layer on the surface of the electrode. The thickness of the raids layers on the AuEC electrode is in the range of 0.778 to 1 μm. X-ray research made it possible to determine the phase state of the raids layers on the investigated electrode. X–ray diffraction lines are from the planes (111, 200, 220, 311) of the following phases: Cu, Cu2O, Au08Cu0.2, and Cu0.88Au0.12.

The analyzed raids layers on a copper electrode with galvanically applied gold consists of two parts differing both in terms of structure and chemical composition—Figure 8. The construction of the part bordering with copper is massive, sometimes discontinuous, while the part bordering with the atmosphere of oxygen and ozone-segmented—Figure 8. The chemical composition of the bordering copper is copper, silver, and gold, and in the bordering part of the atmosphere oxygen, copper, and gold—Figure 8, Table 1.

Metallographic investigation did not reveal microscopic raids layers on the surface of the electrode made of gold. As a result of the microanalysis of the chemical composition using EDX, however, the presence of oxygen in several micro-areas on the surface of this electrode was revealed—Figure 11 and Figure 12.

In the gold structure, very large grains were observed with annealing twins and geometrical figures inside the grains—Figure 9. The average surface size of the grains disclosed is from about 0.25 to 1.15 mm^2^—Figure 9. In addition, observations in the scanning microscope allowed to state that the surface of the golden electrode is characterized by a very large development line—Figure 10.

On a diffractogram made in X-ray research from Au surface there are only diffraction lines originating from the gold planes with indicators (111), (200), (311) and (222) corresponding to the interplanar distances, respectively: d = 2.35Å, 2.04Å, 1.44Å, 1.29Å and 1.17Å—Figure 13. However, quantitative microanalysis from micro-areas of the electrode surface made using EDX showed the presence of oxygen in the experiments micro-areas—Figure 11, Figure 12, Table 2. The research results do not allow to determine the presence of raids layers on the surface of the electrode made of gold. The occurrence of oxygen in micro-areas on the surface of the electrode was found. This is the reason for the slight decrease in the amount of ozone produced. Due to the great ease with which oxygen forms compounds, it can be assumed that in the micro-areas in which the presence of oxygen is revealed, there are probably gold oxides (Au0.6O0.4, Au0.78O0.22)—Table 2, Figure 12.

## 4. Conclusions

In the ozone production process, the plasma reactor electrode works in an aggressive environment of generated ozone. As a result of working in such conditions, the electrode is oxidized in a process of chemical corrosion. The layer of corrosion products created during the work of the plasma reactor isolates the surface of the electrode, which reduces the intensity of the electric field, causing a decrease in the amount of plasma generated, which is a direct cause of the reduced concentration of ozone during this process.

The dynamics of the plasma generation process and the type of electrode material working in the changing process conditions are the decisive factors affecting the concentration of ozone produced. Two electrodes were compared; the surface layer of which was the same, made of gold, and they showed different plasma generation characteristics during operation. The influence of the medium, which is the electrode material, depends mainly on its resistance to corrosion in an environment of dynamically changing conditions, i.e. electrode rotation, oxygen flow through the rotating electric field, and the long one-month working time of the plasma reactor. Corrosion products that formed on the surfaces of individual electrodes after 30 days of work in the plasma reactor showed different geometry, structure, and chemical composition in the micro-areas of the raids layers. The raids layer formed during the operation of the copper electrode with electrolytically-deposited gold is from 1.5 to 2.0 μm and has a double-layer form. The first layer, bordering with copper, is a discontinuous electroplated coating with a thickness of 0.78 μm, and the other, located at the border with the environment, is a layer of fragmented form with a thickness of about 2.0 μm. The microanalysis of the layer bordering with copper showed that its chemical composition is Au, Ag, O and Cu, while in the layer bordering with the environment—oxygen, gold and copper. The formation of the raids layer in this form is the result of a local disappearance of galvanically-deposited gold during the plasma generation process and the deposition of this metal on the walls of the plasma reactor. The electrode surface deprived of the protective layer of gold oxidized. The reason for this behavior of the electrode is the fact that gold used as a coating material with a high positive electrochemical potential protects the electrode surface (copper) only in a mechanical way, not an electrochemical one.

On the surface of the gold electrode, during microscopic and X-ray investigation, no continuous coating was found. However, research using the scanning electron microscope revealed the presence of oxygen in the micro-areas of the electrode surface. In the chemical composition of the analyzed micro-areas, the amount of oxygen is about 10%.

## Figures and Tables

**Figure 1 materials-12-02137-f001:**
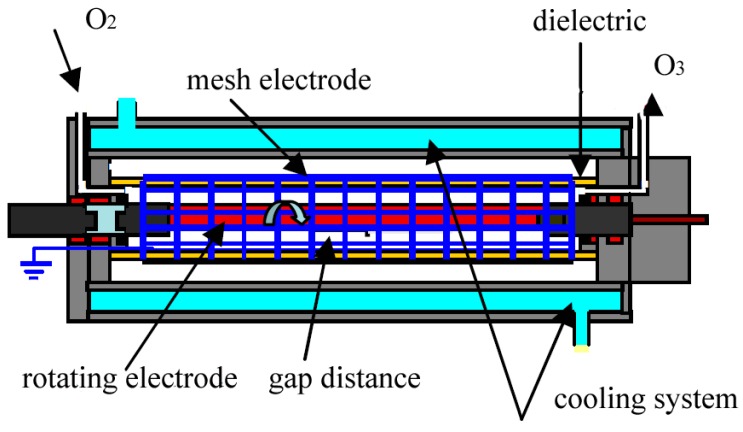
Schematic diagram of plasma reactor with rotating electrode.

**Figure 2 materials-12-02137-f002:**
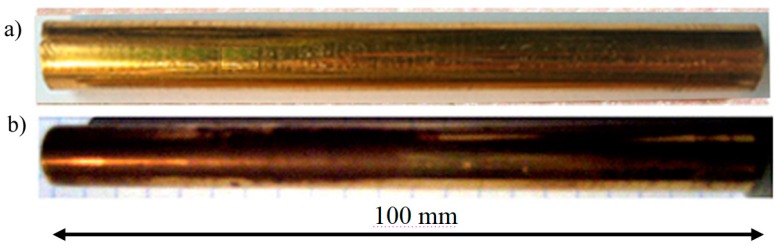
Photograph of electrodes after monthly use in a plasma reactor: (**a**) gold electrode, (**b**) copper electrode galvanized with gold and matte after use.

**Figure 3 materials-12-02137-f003:**
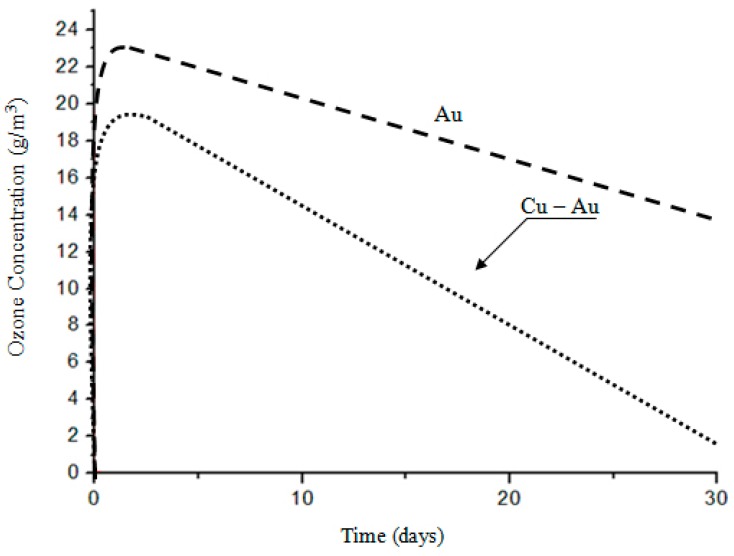
Dependence of the ozone concentration on the working time of electrodes.

**Figure 4 materials-12-02137-f004:**
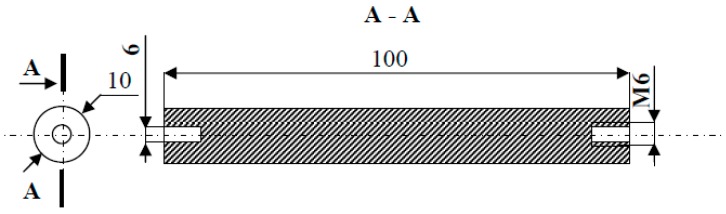
The shape and dimensions of the electrode made of gold.

**Figure 5 materials-12-02137-f005:**
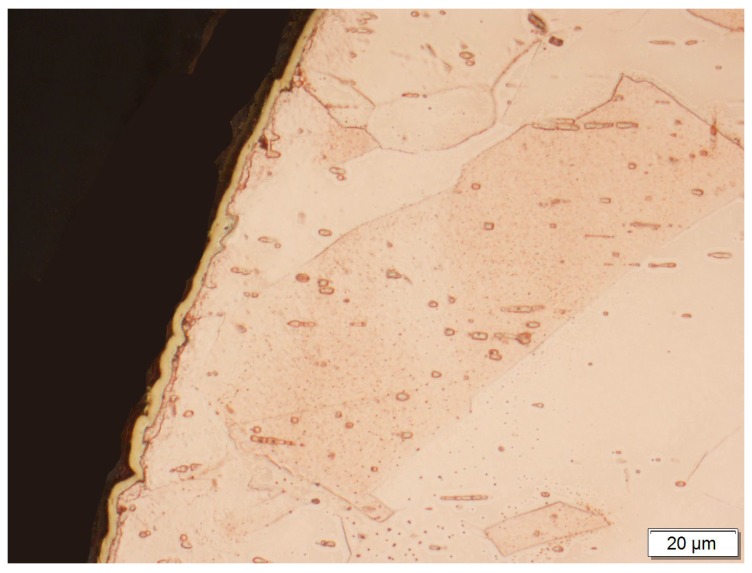
A gold layer applied galvanically to the surface of the copper electrode.

**Figure 6 materials-12-02137-f006:**
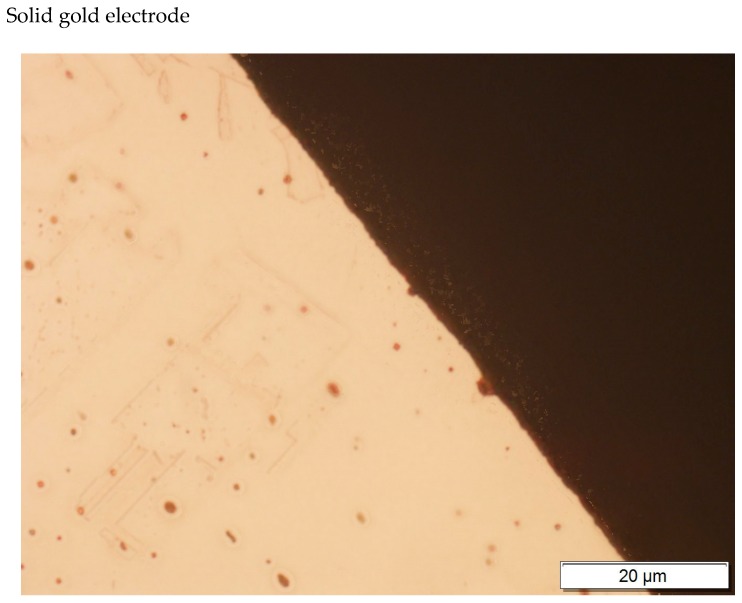
Gold electrode photography.

**Figure 7 materials-12-02137-f007:**
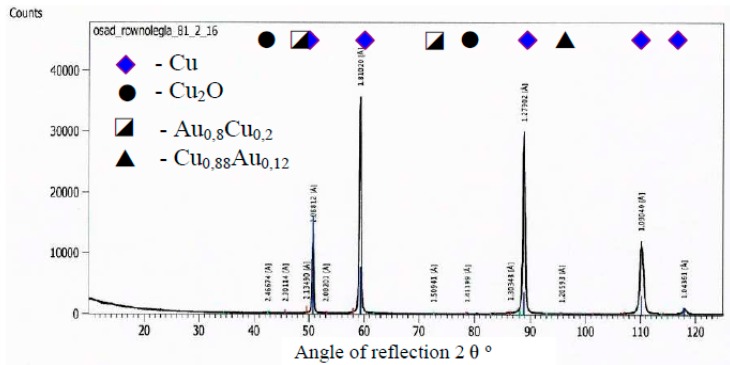
A diffractogram of the surface of a copper electrode galvanized and coated by gold.

**Figure 8 materials-12-02137-f008:**
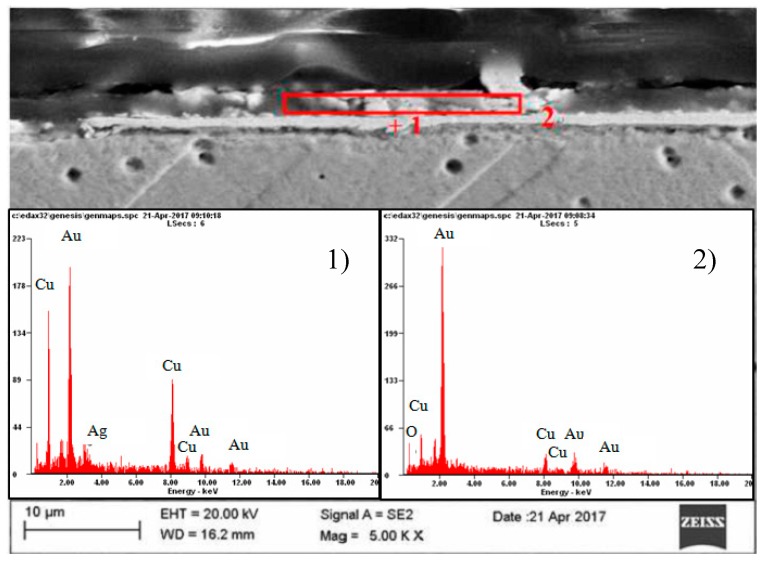
Results of point (**1**) and surface (**2**) microanalysis of the chemical composition of the coating layer on gold electrolytically deposited on copper.

**Figure 9 materials-12-02137-f009:**
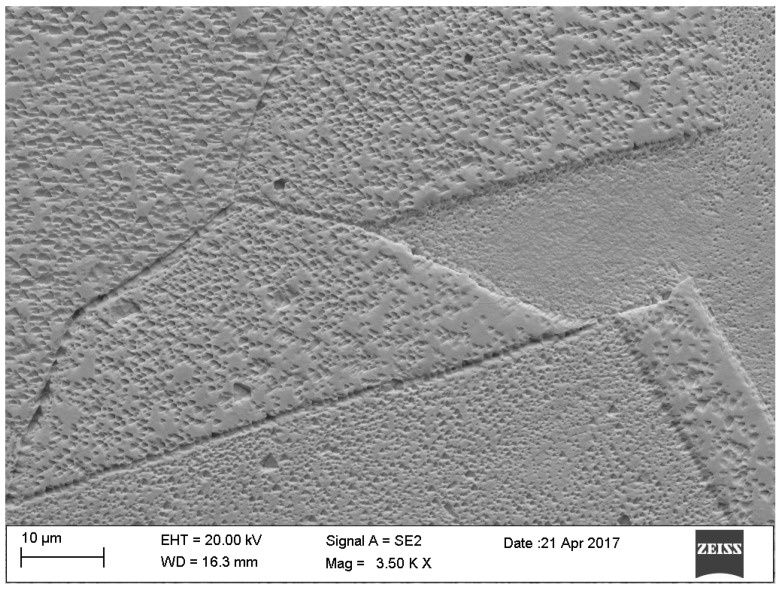
Large Au grains with twins in the structure.

**Figure 10 materials-12-02137-f010:**
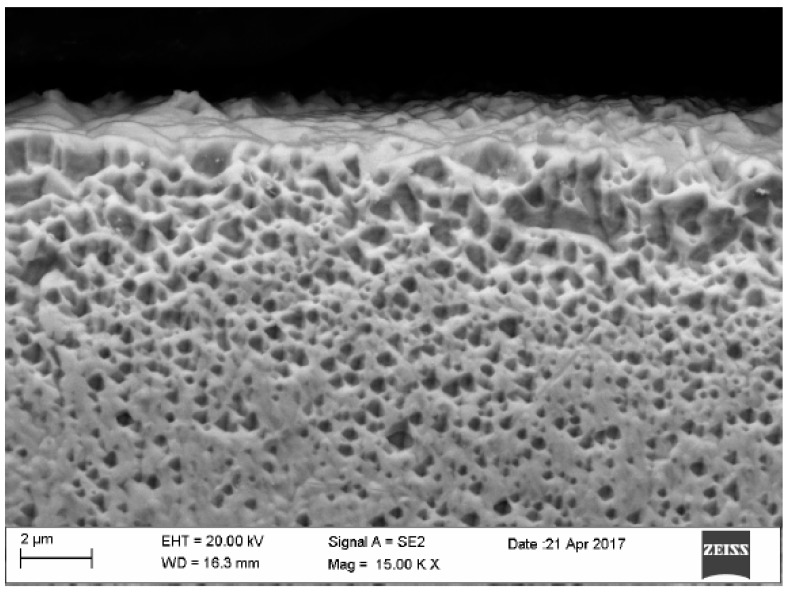
Surface of gold electrode.

**Figure 11 materials-12-02137-f011:**
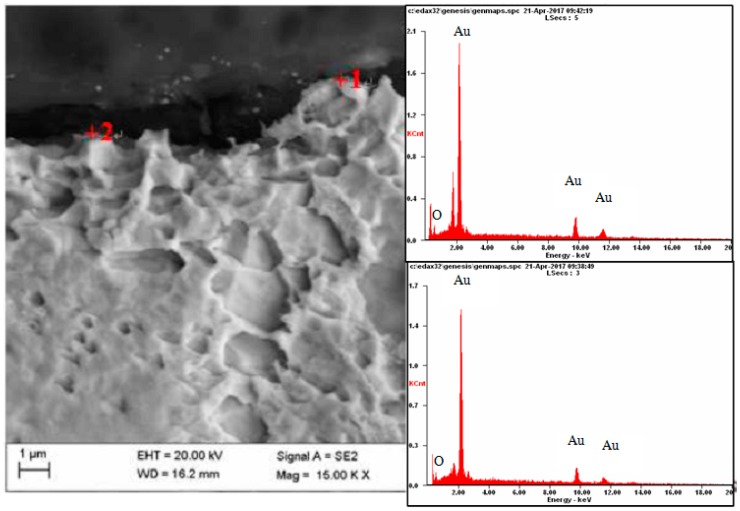
Results of microanalysis of the chemical composition of the Au electrode, from the areas marked in the photomicrographs.

**Figure 12 materials-12-02137-f012:**
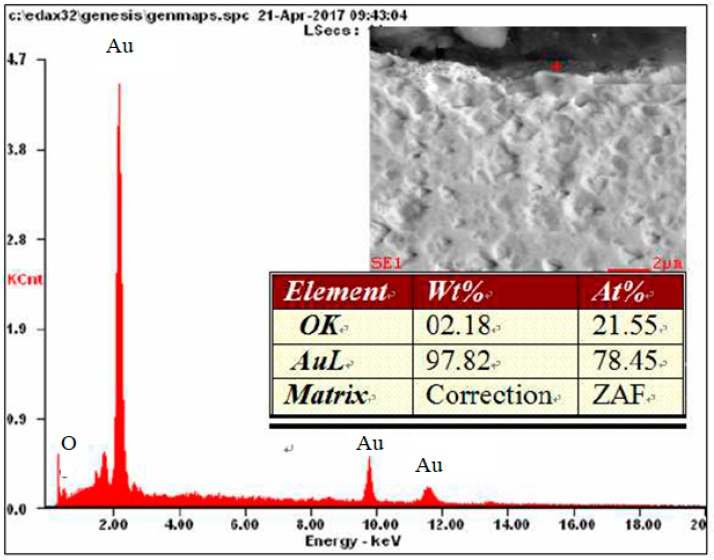
Results of microanalysis of the chemical composition of the Au electrode.

**Figure 13 materials-12-02137-f013:**
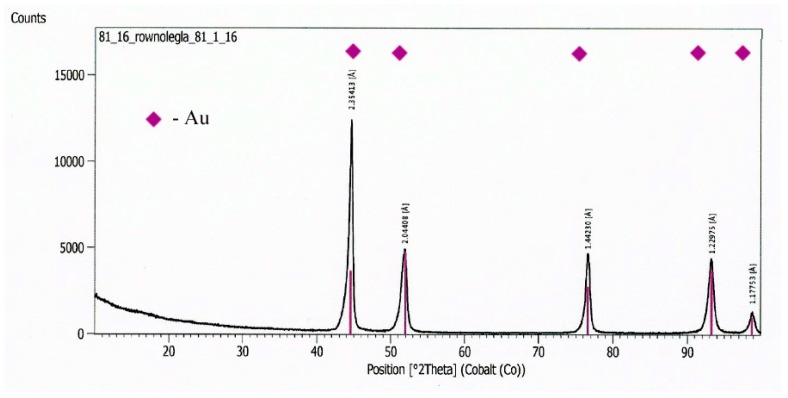
A diffractogram from the Au electrode surface.

**Table 1 materials-12-02137-t001:** Results of microanalysis of the chemical composition of the coating layer on a copper electrode with electrolytically-deposited gold.

Element	Place of Measurement			
	1		2	
	% wag	% at	% wag	% at
OK	04.2	28.7	---	----
CuK	14.9	25.9	38.1	63.3
AuL	80.9	45.4	54.2	29.0
AgL	----	-----	07.7	07.5

**Table 2 materials-12-02137-t002:** Results of the quantitative microanalysis of the chemical composition of the Au electrode surface.

Element	Place of Measurement			
	1		2	
	% wag	% at	% wag	% at
OK	05.1	40.0	02.3	22.4
AuL	94.9	60.0	97.7	77.6

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
