# Peer review of "Investigation of the Condition of the Gold Electrodes Surface in a Plasma Reactor"

_materials, 2019, doi:10.3390/ma12132137_

Reviewer 1 Report

I recommend changing the title as follows: “Investigation of the Condition of the Gold Electrodes Surface in Plasma Reactor”

English should be improved.

This is my recommendation for the abstract:

“During the long-term operation of the plasma reactor, decreases in the plasma concentration were noticed despite the constant maintenance of all parameters. One of the factors is the decrease in the nitrogen content on the electrode surface; in order to eliminate it, the supply voltage was increased up to 11 kV. The next decisive factor in the plasma concentration decrease is the oxidation of the electrode surface. These effects were studied using two electrodes: the gold one and the copper electrode coated with 10 ?m thick layer of the galvanized gold. In the experiment with the gold coated electrode, a large decrease in plasma concentration was observed. High-energy electrons have knocked out the gold atoms from the electrode; as a result, the gold atoms were evaporated and the raids layers formed. After a month - long term of the operation of the electrodes, metallographic analyzes was carried out, the results of which are described in this publication”.

These are only some of my comments concerning English:

1. Ozone concentration was decreased

2. Fig.1 and throughout the text “copper electrode galvanized coated with gold”. Maybe, “copper electrode galvanized with gold” or “copper electrode galvanized with gold” would be sufficient.

3. Figure 2. Dependence of the ozone concentration on the working time of electrodes.

4. The first sentence in “3. Results” I recommend: “At first, the gold layer at the copper electrode surface was investigated”. Comment: it is not necessary to repeat many times that the layer was galvanized.

And so on throughout the text…

Please, along with [3,7], add a citation expedient to the “Einstein's theory”.

Very important comment: Many figures are not clear, with diffractograms especially. I recommend adding inscriptions.

Author Response

Dear Reviewer,

Thank you so much for your help and kindness.

Everything has been improved according to your recommendations.

I recommend changing the title as follows: “Investigation of the Condition of the Gold Electrodes Surface in Plasma Reactor”

The title has been changed

English should be improved.

English has been improved.

This is my recommendation for the abstract:

“During the long-term operation of the plasma reactor, decreases in the plasma concentration were noticed despite the constant maintenance of all parameters. One of the factors is the decrease in the nitrogen content on the electrode surface; in order to eliminate it, the supply voltage was increased up to 11 kV. The next decisive factor in the plasma concentration decrease is the oxidation of the electrode surface. These effects were studied using two electrodes: the gold one and the copper electrode coated with 10 ?m thick layer of the galvanized gold. In the experiment with the gold coated electrode, a large decrease in plasma concentration was observed. High-energy electrons have knocked out the gold atoms from the electrode; as a result, the gold atoms were evaporated and the raids layers formed. After a month - long term of the operation of the electrodes, metallographic analyzes was carried out, the results of which are described in this publication”.

Abstract is changed.

These are only some of my comments concerning English:

1. Ozone concentration was decreased

2. Fig.1 and throughout the text “copper electrode galvanized coated with gold”. Maybe, “copper electrode galvanized with gold” or “copper electrode galvanized with gold” would be sufficient.

3. Figure 2. Dependence of the ozone concentration on the working time of electrodes.

4. The first sentence in “3. Results” I recommend: “At first, the gold layer at the copper electrode surface was investigated”. Comment: it is not necessary to repeat many times that the layer was galvanized.

And so on throughout the text…

Descriptions of drawings and other English mistake have been corrected.

Please, along with [3,7], add a citation expedient to the “Einstein's theory”.

The citation papers about energy and Einstein's theory are added in references 1, 23, 25,

Very important comment: Many figures are not clear, with diffractograms especially. I recommend adding inscriptions.

Figures and diffractograms have been changed to a better quality and inscriptions have been added

Reviewer 2 Report

The authors of this paper presented an interesting research. The paper is accepted for publication if authors will answer the following comments:

1) What is the originality of the paper?

2) There is little data about the experimental conditions of the plasma reactor. A figure with the plasma reactor should be added.

3) Throughout the paper some data is written with point as decimal separator and some data with comma.

4) Number of references are not enough, please refer the surface treatment by plasma, as well as the electrode deterioration.

Author Response

Dear Reviewer,

Thank you so much for your help and kindness.

Everything has been improved according to your recommendations.

According first Reviewer suggestion:

The title has been changed  “Investigation of the Condition of the Gold Electrodes Surface in Plasma Reactor”

Abstract is changed.

English has been improved.

Descriptions of drawings and other English mistake have been changed.

The citation papers about energy and Einstein's theory are added in references

Figures and diffractograms have been changed to a better quality and inscriptions have been added

I will try to answer your questions.

1) What is the originality of the paper?

 The original of this paper are:

showing that the use of a plasma reactor electrode made of gold improves the efficiency of ozone generation. Ozone production is becoming stable. The next important information is that when we use a copper electrode galvanized with gold, gold is selected by highly energetic electrons and the efficiency of the plasma reactor decreases with operating time. The conclusion is that a galvanized gold electrode can not be used because after long-lasting continuous operation, the gold layer will be struck by the electrons of the plasmas. No one has done research on a gold-plated and gold-plated electrode before.

2) There is little data about the experimental conditions of the plasma reactor. A figure with the plasma reactor should be added.

 Data about the experimental conditions of the plasma reactor are added in Introduction page number 1 and a figure the plasma reactor is added (Figure 1).

The rotating electrode was coated with gold and high voltage was applied on it. The speed of the rotating electrode was varied from 0 to 800 rpm by a variable speed motor. The dielectric barrier covered by a mesh electrode was a glass tube of length 110 mm, diameter 15 mm and its thickness was 1.25 mm. The outer mesh electrode made from copper wires with diameter of 0.1 mm was grounded. The size of the copper mesh electrode was 0.2 mm square. 99.5 % oxygen gas regulated by a digital mass flow controller was fed at a gas flow rate ranging from 0.5 l/min to 2 l/min. Applied voltage and its frequency were set at 9 - 10 kV and about 12 kHz, respectively. Figure 1 shows schematic diagram of our ozonizer. The discharge gap distance was 1.1 mm and the discharge length along the reactor was 100 mm. The gas temperature at the outlet of the ozonizer was measured during the experiments. The measured data was saved every day in a computer. All experiments were carried out at atmospheric pressure in oxygen at around room temperature (15 ~ 30°C). In our case cooling water temperature was 10°C.

3) Throughout the paper some data is written with point as decimal separator and some data with comma.

 Thank you for seeing this mistake. Now have been corrected

4) Number of references are not enough, please refer the surface treatment by plasma, as well as the electrode deterioration.

Number of references are addend.

I have increased the number of references from 12 to 30 papers according your suggestions refer the surface treatment by plasma, as well as the electrode condition.

Reviewer 3 Report

The manuscript "Investigation of Gold Electrodes Surface Condition of Plasma Reactor"  describes the poisoning of the gold electrodes and gaves some hints how to reduce the negative side-effects leading to deacrease of the ozone production. This manuscript is too technical and lacks of scientific soundness for the MATERIALS readers. I consider that this paper can be re-submitted to another MDPI journal as it is out of scope of MATERIALS due to the lack of novelty and scinfic results.

Author Response

Dear Reviewer,

Thank you so much for checking our paper.

The manuscript not describes the poisoning of the gold electrodes and did not gave some hints how to reduce the negative side-effects leading to deacrease of the ozone production. This manuscript describes how is change surface condition during ozone generation. Paper describes oxydation effect and it is reson why was send to MDPI Metals.

Have a good day,

Author

Everything has been improved according to reviewers recommendations.

Round  2

Reviewer 2 Report

The authors of this paper replied to the reviewer’s questions thus the manuscript is accepted for publication.

Author Response

Dear Reviewer,

Thank you very much for accepting my paper.

Two native English teachers checked my paper again.

Best regards,

Sebastian

Reviewer 3 Report

First of all, I recommend to authors to pay attention what journal they are submitting the paper. The manuscript is submitted to MATERIALS, and my key point was that this not a perfect match with the audience of Materials. In the response to my suggestion authors wrote : "Paper describes oxydation effect and it is reson why was send to MDPI Metals."

So dear authors, what journal you want to submit this paper? I agree, that Metals would be better choice. So please double check.

Then, I want hesitate to check your microanalysis results. The use of second decimal in this case is not appropriate because the precision of this analysis is 0.1 at.% (at best).  Please round these value to first decimal. In addition authors could provide the standard deviation. Regarding the oxygen content, it might be useful to check Oxygen concentration by XPS.

In the original version the mistakes were very abundant (e.g. rector instead of reactor), now it is much better, but please check the final version once again.

Author Response

Dear Reviewer,

Thank you so much for checking, suggestions and advice. Two native English teachers checked our paper. I rounded the value to the first decimal place. You suggest that  it might be useful to check oxygen concentration by XPS. Surely you are right to be reliable if it was done right after the experiment, but we did not think about it. I hope that you will allow us to publish this paper and in the future students will improve the quality of their research and papers.

Best regards,

Sebastian
